# Detection of SARS-CoV-2 Using the Abbott™ PANBIO™ COVID-19 SELF-TEST Rapid Test in Patients Seen at INER

**DOI:** 10.3390/biomedicines13051012

**Published:** 2025-04-22

**Authors:** Kenny Alonso Cantón Cruz, Martha Angella Durán Barrón, Israel Agustín Morales Lozada, Mario Alberto Mujica Sánchez, Grecia Gabriela Deloya Brito, María del Carmen García Colín, Hansel Hugo Chávez Morales, José Nicolás Aguirre Pineda, Cinthya Karen Cid del Prado Rojas, Stephanie Jara Valdés, Eduardo Becerril Vargas

**Affiliations:** Laboratory of Clinical Microbiology, National Institute of Respiratory Diseases (INER), Mexico City 14080, Mexico; kennycantoncruz@gmail.com (K.A.C.C.); m.angybarron@gmail.com (M.A.D.B.); israel_agustin_ml@outlook.com (I.A.M.L.); mario.mujica1@gmail.com (M.A.M.S.); mcgarcia09@hotmail.com (M.d.C.G.C.); qbp_hhcm@hotmail.com (H.H.C.M.); nicolas.aguirre3091@gmail.com (J.N.A.P.); cid_karen@hotmail.com (C.K.C.d.P.R.); fanyjaravaldes@gmail.com (S.J.V.); edobec.var@gmail.com (E.B.V.)

**Keywords:** coronavirus, SARS-CoV-2, COVID-19, Panbio™ COVID-19 antigen self-test

## Abstract

The COVID-19 pandemic has highlighted the need for rapid and accurate tests to detect SARS-CoV-2. **Objectives:** This study evaluates the effectiveness of the Panbio™ COVID-19 Antigen Self-Test rapid test compared to reverse transcriptase polymerase chain reaction (RT-PCR). **Methods**: A prospective diagnostic testing study was performed. Patients with respiratory symptoms who provided informed consent were included. **Results:** We included 205 patients with COVID-19 symptoms who underwent both tests. The mean age was 35.55 ± 12.62 years, and 64% of the participants were female. Sensitivity and specificity were 71.2% (95% CI: 62.5–79.9%) and 100% (95% CI: 96.4–100%), respectively. **Conclusions:** If a test is positive within the first 72 h after the onset of symptoms, we can be sure that it is a case of COVID-19; however, when the antigen test is negative, confirmation with RT-PCR is necessary. Its ease of use and results with moderate precision make it a valuable tool for early detection.

## 1. Introduction

The COVID-19 pandemic highlighted the importance of diagnostic testing to limit mortality and viral transmission. These tests have advanced viable therapeutic interventions and enabled the development of healthcare strategies to control the spread of the infection, a need that persists as the global health emergency enters an endemic phase [1]. Polymerase chain reaction (RT-PCR) remains the gold standard for SARS-CoV-2 detection. However, its limitations, including high costs, longer turnaround times, the need for specialized personnel, and the risk of infection at testing sites, have prompted the development of rapid antigen localization tests (RADTs) as a viable alternative [2]. Among RADTs, self-testing has emerged as a transformative tool, enabling individuals to test at home and advancing access to testing, particularly in resource-limited or isolated settings. These tests, including the widely used Abbott™ PANBIO™ COVID-19 Ag Rapid Test (a precursor to the self-test version), detect viral proteins such as the nucleocapsid (N) antigen in minutes without requiring advanced equipment or trained personnel [3]. The PANBIO rapid assay, typically administered by healthcare professionals, shares the same lateral flow immunoassay technology as the self-test evaluated here, differing primarily in its administration mode: the self-test is designed for home use by untrained individuals, enhancing accessibility but potentially introducing variability in sample collection [4]. Some studies have been published indicating that self-administered tests increase the reach of testing, reduce the risk of transmission by minimizing contact in healthcare facilities, and facilitate early isolation, key factors in controlling episodes [5,6].

For example, a pilot study demonstrated that weekly self-testing in high-risk populations, such as healthcare workers and school staff, was feasible and effective in rapidly detecting cases, contributing to control efforts [7]. Furthermore, self-testing is cost-effective and rapid, taking only 15 min, allowing for more testing per visit to compensate for its lower sensitivity compared to RT-PCR [8]. This frequency is particularly advantageous, as viral loads peak 2–3 days after symptom onset, coinciding with the period of highest contagiousness [9]. By shifting the responsibility of testing to patients, self-administered testing reduces the burden on the healthcare system, expands the reach of comprehensive care, and facilitates discreet risk-reduction decision-making [10]. However, the performance of self-administered tests can vary across SARS-CoV-2 variants, a problem exacerbated by the continued spread of the infection. Variants such as Omicron and its sublineages exhibit changes in target proteins that can reduce the sensitivity of antigen-based tests, potentially increasing false-negative rates [11,12]. Research on the use of deep mutational screening has identified specific changes in the nucleocapsid that inhibit the efficacy of specific antigens in symptomatic testing, highlighting the need for ongoing evaluation to ensure stable quality against emerging strains [13]. Despite these concerns, meta-analyses show that self-administered rapid-antigen diagnostic tests maintain satisfactory accuracy when used correctly. The highest sensitivities described for symptomatic people were 73.91%. This study did not include an evaluation of the performance of the Abbott™ PANBIO™ COVID-19 self-test, the test evaluated in this study [14].

The Panbio™ COVID-19 Antigen Self-Test is a lateral flow immunoassay intended for the qualitative detection of SARS-CoV-2 nucleocapsid antigens in nasal swabs, designed for use by individuals without medical training, making it accessible for home use. Evaluating the performance of such self-tests is crucial to ensure their reliability in detecting infections, particularly in the context of public health strategies aimed at controlling the spread of COVID-19. While manufacturer-reported sensitivities and specificities are available, real-world performance may differ due to variations in user technique, sample collection, and interpretation of results. Therefore, independent evaluations in clinical settings are necessary to validate these tests. Despite its wide adoption—evidenced by more than 10 million targeted and self-test cases in the United States between October 2021 and June 2022—few studies have comprehensively evaluated its performance, efficacy, and epidemiological impact in clinical practice [15]. This gap is notable, as the predecessor PANBIO rapid assay has been extensively studied, with sensitivities of 82.5% to 88.98% and specificities near 100% in symptomatic populations [14,15]. At the same time, evaluations of the self-test are limited, though Cai et al. (2024) report a pooled sensitivity of 73.91% for self-performed antigen tests [14]. While the professionally administered Panbio™ Ag test has been extensively studied, published data on the real-world performance and user experience specifically for the self-test version remains relatively less common compared to its widespread use. Independent evaluations like ours are therefore valuable in confirming their utility and limitations when performed by untrained individuals in diverse settings. This study addresses this need by evaluating the real-world performance of the Abbott™ PANBIO™ COVID-19 self-test compared to RT-PCR in symptomatic patients at INER.

## 2. Materials and Methods

### 2.1. Study Design and Participants

We conducted a prospective, cross-sectional diagnostic test study. We included all patients referred to the Clinical Microbiology Laboratory at the National Institute of Respiratory Diseases (INER), Mexico City, Mexico, with respiratory symptoms, who were requested to undergo pharyngeal and nasopharyngeal sampling for PCR testing of influenza, SARS-CoV-2, and other respiratory viruses between May 2022 and October 2022. All patients were invited to participate in the study. Patients who agreed to participate signed an informed consent form and received a box containing a test device, a sterile swab, a vial with buffer solution, a test tube with a blue cap, and a tube holder. The box included a quick reference guide with illustrated instructions for sample collection and processing. After reviewing the kit guide, participants swabbed both nostrils, processed the test, and interpreted the results according to the instructions. A trained observer witnessed and evaluated the process without intervening. At the end, participants verbally communicated the result to the observer, who documented and photographed it. The participants then entered a second cubicle to have a sample taken for molecular study. The samples obtained were placed in a tube with a universal viral transport medium to preserve the genetic material until analysis in molecular biology.

### 2.2. Self-Test Procedure

Every participant performed the Panbio™ COVID-19 Antigen Self-Test (Abbott Rapid Diagnostics, Jena, Germany) following the manufacturer’s instructions provided in the kit. The test involves collecting a nasal swab sample, processing it with the provided buffer solution, and applying it to the test device. Results were read after 15 min, with the appearance of a control line indicating a valid test and a test line indicating a positive result. A trained observer witnessed the process without intervening and recorded the participant’s interpretation of the result. For detailed procedures, refer to the manufacturer’s protocol (https://globalpointofcare.eifu.abbott/es-la/detail-screen.html, accessed on 19 April 2025).

### 2.3. RT-PCR Testing

SARS-CoV-2 RNA detection in nasopharyngeal specimens followed the national guidelines of the National Institute for Epidemiological Reference (InDRE) [11]. At INER, samples were collected with pharyngeal and nasopharyngeal swabs and preserved in 3 mL of viral transport medium (UTM). These samples underwent an automated nucleic acid extraction process on Class II biosafety equipment in a biosafety level 2 facility. RNA extraction was performed from 200 μL of oropharyngeal/nasopharyngeal exudate samples in a universal transport medium, using the BIONEER Exiprep 96 kit and the ExipPrep 96 Viral DNA/RNA extraction kit (Ref. K-4614, Bioneer Corporation, Daejeon, South Korea), following the manufacturer’s specifications. RT-PCR was conducted for viral RNA amplification using the GeneFinder™ COVID-19 Plus RealAmp kit (Ref. IFMR-45, Osang Healthcare Co., Ltd., Anyang, South Korea), which amplifies the *RdRP, N, and E* genes. A total volume of 20 μL was prepared by mixing 10 μL of the master mix, 5 μL of the probe mix, and 5 μL of the nucleic acid extract. RT-qPCR was carried out on a Quant Studio 5 thermal cycler (Applied Biosystems, Thermo Fisher Scientific, Waltham, MA, USA) under the following conditions: 50 °C for 20 min, 95 °C for 5 min, followed by 45 cycles of 95 °C for 15 s and 58 °C for 60 s.

### 2.4. Detection of Other Respiratory Viruses Using the Luminex System

The detection of other respiratory viruses was performed using the NxTAG^®^ Respiratory Pathogen Panel (Luminex Corporation, Austin, TX, USA, Cat. No. RPP-1), which includes reagents to detect 19 viral types and subtypes. Bacteriophage lambda was included in each run to control amplification and assay performance. The assay comprised reverse transcription followed by multiplex PCR amplification and a bead hybridization step, performed according to the manufacturer’s instructions. Signals presented as mean fluorescence intensity (MFI) were acquired on the Luminex 200 platform by flow cytometry.

### 2.5. Statistical Analysis

Patient demographic and clinical information were obtained from the National Epidemiological Surveillance System. A database was created with the results obtained from the processing of the samples by both methods. Statistical analysis was performed using SPSS 24.0 (IBM, Armonk, NY, USA). Means with standard deviations were calculated for quantitative variables, and frequencies and percentages were reported for qualitative variables. Patients were divided into three groups: patients with COVID-19, patients with symptoms but negative tests for all viruses, and patients with identification of other respiratory viruses. The corresponding results were compared using the Cochran-Mantel-Haenszel test. For the analysis of qualitative variables, a student’s t-test and/or Mann-Whitney U test was performed, depending on whether the variables were parametric or nonparametric. *p*-values for multiple comparisons of clinical characteristics were adjusted using the Bonferroni correction. The value of *p* < 0.05 was considered statistically significant.

The clinical and demographic variables of patients with COVID-19 and patients with other respiratory viruses were compared, and the diagnostic performance of the Panbio™ COVID-19 Antigen Self-Test was evaluated in comparison to the diagnostic standard, RT-PCR. Using these data, the following parameters were calculated: sensitivity (true positive results divided by the sum of true positive and false negative results), specificity (true negative results divided by the sum of true negative and false positive results), positive predictive value (PPV, true positive divided by the sum of true positive and false positive results), negative predictive value (NPV, true negative divided by the sum of true negative and false negative results), and negative likelihood ratio (LR-, 1-sensitivity divided by specificity). The concordance between tests was assessed using Cohen’s kappa coefficient. Subgroup analyses were conducted to evaluate the influence of age, sex, and symptom severity on test performance.

This study follows the Declaration of Helsinki and was approved by the Research Ethics Committee of the National Institute of Respiratory Diseases (INER) under study no. C04-23. Informed consent was obtained from all participants before they participated in the study.

## 3. Results

A total of 205 patients were included, with a mean age of 35.55 ± 12.62 years. Sixty-four percent (132/205) of the participants were women (Figure 1). The higher proportion of female participants may reflect a greater tendency among women to seek medical attention for respiratory symptoms, though this was not specifically assessed in this study.

Of the patients evaluated, 42.4% (87/205) had at least one comorbidity. The most common were overweight or obesity (52%), followed by arterial hypertension (21%) (Table 1).

SARS-CoV-2 infection was confirmed in 104 patients by a positive RT-PCR result. Of the patients included in the study, 81 had a negative molecular test for SARS-CoV-2 and other respiratory viruses, and 20 were reported positive for other respiratory viruses, the most frequent being enteroviruses/rhinoviruses, coronavirus OC43, and influenza A/B (Figure 2). The most frequent symptoms in the patients tested were rhinorrhea, cough, and headache (Table 2). The median symptom onset was 1.60 (± 1.85) days. No differences were found in the symptoms reported by patients with COVID-19 compared to those caused by other infectious diseases caused by other respiratory viruses. None of the patients received antivirals or steroids prior to the study. Forty-two percent took symptomatic treatment with NSAIDs, and only four patients used antibiotics prior to testing (Table 2).

Using the Abbott™ PANBIO™ COVID-19 SELF-TEST, 74 tests were positive for SARS-CoV-2. None of the samples obtained for antigen testing were invalid for reading during the study. The sensitivity of the rapid antigen test was 71.2% (95% CI: 62.5–79.9%), and specificity was 100% (95% CI: 96.4–100%), with a kappa index of 0.71 (Table 3).

Better diagnostic performance was observed in patients tested 48 h after symptom onset. A specificity of 100% and 93% sensitivity 2 days after symptom onset were obtained in those tested (Table 4).

The Ct (cycle threshold) values for the rapid test-negative and PCR-positive samples are detailed in Table 5. The average Ct for the *E, N, and RdRP genes* from the GeneFinder™ kit was lower in the rapid test-positive patient samples compared with the average Ct of the negative samples (Figure 3).

Subgroup analysis was performed to assess the impact of age, sex, and being a health care professional on the sensitivity of the Panbio™ COVID-19 Antigen Self-Test. Multivariate regression analysis identified no statistically significant predictors of test sensitivity among these factors (*p* > 0.05 for all). (Table 6 and Table 7)

## 4. Discussion

Despite the termination of the international public health emergency for pandemic SARS-CoV-2 on 5 May 2023, after three years and two months, this does not indicate that the disease is no longer a global threat due to reinfections associated with the emergence of circulating variants of SARS-CoV-2 that can evade the immune response [9]. Therefore, widespread testing remains important to identify symptomatic and asymptomatic individuals, provide timely medical treatment, and reduce transmission [10,11]. Self-testing is an option for individuals seeking accessible testing. In the United States, from October 2021 to May 2022, 10.7 million voluntarily reported self-test results were recorded [12]. Other countries, such as Singapore, have implemented self-testing as a new standard when the nation reopened its borders to the world in April 2022 [13]. Despite its large-scale implementation and use, there is still a lack of knowledge about its performance, acceptability, and impact on epidemiological surveillance due to the lack of mandatory reporting of the results by users.

This study compared the performance of the rapid Panbio™ COVID-19 Antigen Self-Test against the gold standard, RT-PCR, at the National Institute of Respiratory Diseases. The results presented in this paper show that Abbott’s PANBIO COVID-19 Ag rapid test for diagnosing SARS-CoV-2 infection with mild symptomatology has high specificity and variable sensitivity depending on the time since symptom onset. The highest performance for detecting SARS-CoV-2 infection is found 48 h after symptom onset, and performance is low when individuals have three or more days of symptom onset. We observed that sensitivity drops significantly in persons who sought medical consultation and diagnostic testing 3 days after symptom onset. The sensitivity observed in people with more than four days of symptoms was 58%. The data obtained on the diagnostic performance for COVID-19 confirmation in our study are comparable to those reported in a systematic review and meta-analysis that included seven articles reporting on the performance of the Abbott Panbio COVID-19 Ag rapid test. Performed by the same patient and without the need for intervention by healthcare personnel, all studies reported a high specificity, between 95.45% and 100%, and a sensitivity ranging from 66% to 88.98%, the latter being 71.2% in our study, using the Panbio self-test kit [14]. For instance, Gertler et al. (2021) reported a sensitivity of 82.5% and specificity of 99.8% in a similar symptomatic outpatient setting [4]. However, their study included professional collection in some cases, unlike our fully self-administered approach [4]. Our study’s unique contribution lies in its evaluation of the self-test in a real-world clinical setting with an observer ensuring procedural accuracy without intervention, providing insights into its practical utility for symptomatic patients.

In 20 patients, other respiratory viruses were detected, six had a seasonal coronavirus as the cause of their symptoms, Coronavirus OC43, which was the most common, all the antigen tests carried out on these 20 patients are negative, data that demonstrate the low or null possibility of having a positive self-testsresult in the presence of other respiratory viruses. The mean Ct values of false-negative results obtained with the test kit under evaluation are higher than those of self-tests with a positive result for SARS-CoV-2. A correlation was found between the level of viral load and the positivity of the self-tests; that is, a low RT-PCR cycle threshold reflects a high viral load, and better sensitivity of RADTs is obtained. Antigen tests can detect infections at an early stage of the disease, between 5 and 7 days after the onset of symptoms. It has been estimated that the viral load is highest during the first 2 to 3 days, a situation that explains the results obtained in our evaluation, in which patients who were tested two days after the onset of symptoms had a sensitivity of 93% [4]. The strong correlation with lower RT-PCR Ct values (Table 5) indicates false negatives at moderate loads (Ct 20-21). These findings largely coincide with previous studies showing lower sensitivity of RATs for Omicron detection compared to Delta when using Ct values as a comparator, especially in samples with low RNA concentrations. Other authors report that Omicron samples present lower ratios of antigen/RNA, which offers a potential explanation for the apparent lower sensitivity of RATs for that variant when using CT value as a reference [16]. Nevertheless, CT values are semiquantitative and can be affected by different factors, such as the patient’s immune status, collection time, and collection method. Therefore, a limit of detection based on a CT value cannot be reliably generalized.

This study evaluated the clinical performance of the self-test during a period (May–October 2022) when the Omicron BA.4 and particularly BA.5 sublineages became dominant in Mexico, displacing earlier BA.2 variants [17]. However, we did not perform genomic sequencing on individual samples to determine the specific variant in each case or to directly assess the impact of specific mutations on the Panbio™ self-test’s sensitivity. Evaluating test performance against specific, characterized variants remains an important consideration for ongoing monitoring. According to a recent study, the sensitivity of rapid antigen tests decreased over time, from 80% to 72% [18]. While the N protein is relatively conserved, mutations at certain sites are still observed. One study showed that the R203K/G204R mutation at the N-terminus of SARS-CoV-2 has been linked to increased infectivity, fitness, and virulence [19]. Whether these changes affect the diagnosis and impact of real-time PCR (RT-PCR) and RADs remains to be determined. In 2021, Lesbon demonstrated the existence of mutations in the *N* gene that could affect the use of SARS-CoV-2 real-time RT-PCR diagnostic kits, influencing false-negative results. These results provide further evidence that existing SARS-CoV-2 variants might escape molecular detection based on nucleic acid amplification tests, especially those using a single viral target [20]. In 2022, Abbott’s Global Surveillance program, which monitors variants and their impact on the performance of COVID-19 diagnostic tests, conducted a study evaluating the performance of molecular, antigen, and serological assays with the following variants (VOC): B.1.1.7 (alpha), B.1.351 (beta), P.1 (gamma), and B.1.617.2 (delta). No decrease in diagnostic capacity was observed with the analyzed variants [21]. Sakai-Tagawa et al. (2023) demonstrated that despite amino acid substitutions in the N protein (P13L, G30F, del31/33, E136D, R203K, G204R or S413R) of several Ómicron variants and their subvariants (BA.5, BA.2.75, BF.7, XBB.1 and BQ.1.1) the sensitivity of rapid tests does not decrease [22]. Similarly, mutations like N_323/K:53, located in the Nucleocapsid (N) protein, were not specifically studied here using functional assays like reverse genetics or protein binding experiments. However, mutations in the N protein can potentially alter antigen recognition by diagnostic tests or antibodies. Furthermore, given the N protein’s role in the viral life cycle, such mutations could theoretically impact viral replication efficiency or contribute to host immune escape mechanisms. However, specific functional studies are needed to confirm these potential effects for the N_323/K:53 mutation. Our 71.2% sensitivity aligns with these ranges, but specific testing against XDV, JN.1.1, or other subvariants is needed [23].

None of the samples obtained for the performance of the Ag test was invalid for reading, which allows for a reliable evaluation of the clarity of the instructions for performing and interpreting the test. This is one of the strengths of the study since there was always an observer who did not intervene in the processing of the test. Therefore, it can be ruled out that the sensitivity was increased with the intervention of a professional for the processing of RADT. It is ruled out that the number of false positives is related to errors in the self-sampling and/or self-testing procedures. Of all the participants, 44.2% were healthcare workers (HCWs) (Table 6). INER has a program for healthcare workers, and all those who come with respiratory symptoms are given a PCR test to detect SARS-CoV-2, Flu, and other respiratory viruses [24]. The multivariate analysis ruled out demographic and social factors that interfered with test performance [16].

Several limitations affect this study. We excluded asymptomatic subjects, limiting assessment of real-world performance in screening populations with potentially lower viral loads and higher false-negative rates (Table 5). The small sample of other respiratory viruses (n = 20), including six seasonal coronaviruses, restricts specificity evaluation against related pathogens, despite no false positives. Cross-reactivity with MERS-CoV or SARS-CoV remains untested [17]. No user satisfaction survey was conducted, leaving unclear whether the observer’s presence influenced adherence, potentially overestimating performance compared to unsupervised use.

## 5. Conclusions

The Abbott™ PANBIO™ COVID-19 SELF-TEST had good specificity but moderate sensitivity. When the antigen test is positive, we can be certain that it is a case of COVID-19; however, when the antigen test is negative, RT-PCR confirmation is necessary. The results obtained in this study demonstrate that antigen testing does not necessarily need to be performed by previously trained healthcare personnel. The results are reliable when patients follow the manufacturer’s instructions, perform the test, process the test, and interpret it. Self-testing offers an alternative to expanding testing coverage and allows for individual decision-making and risk reduction. However, a strategy for reporting results to public health authorities would need to be implemented, as it could lead to a lack of accuracy in the total case count and detract from the impact of COVID-19 when there are outbreaks or an increase in community cases. To understand the impact of the mutations that SARS-CoV-2 has accumulated and that have given rise to the different variants, it is important to continue conducting studies that corroborate the performance of currently available diagnostic tests. In addition to mutations, it is important to consider many other factors, such as the immunization status of patients, which can also have a deleterious impact on the ability to confirm COVID-19 cases.

## Figures and Tables

**Figure 1 biomedicines-13-01012-f001:**
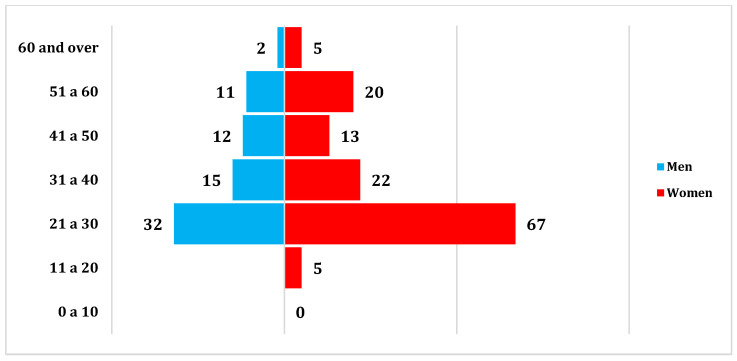
Distribution by sex and age.

**Figure 2 biomedicines-13-01012-f002:**
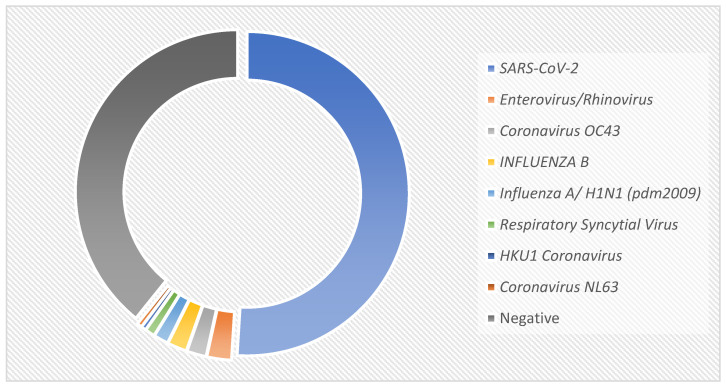
Etiology of respiratory infections.

**Figure 3 biomedicines-13-01012-f003:**
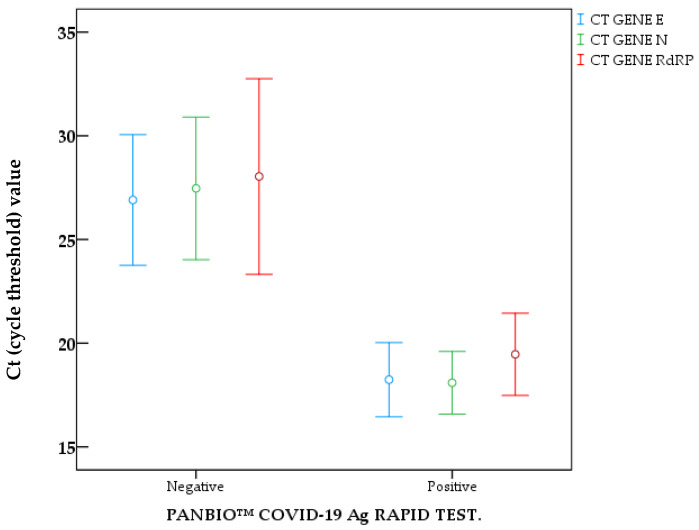
Median CT-PCR RT of E, N, RdRP genes compared to positive and negative tests with PANBIO™_COVID-19_Ag_RAPID_TEST.

**Table 1 biomedicines-13-01012-t001:** The frequency of comorbidities in patients included in the analysis.

Comorbidities	N (87)	%
Overweight or Obesity	45	52%
HTA	18	21%
Asthma	12	14%
DM	8	9%
Cardiopathy	3	3%
ERC	2	2%
Other	26	30%

**Table 2 biomedicines-13-01012-t002:** Clinical characteristics.

	Total	Patients with COVID 19	Patients with Other Viruses	Patients with Negative PCR	*p*	OR	IC
	(N = 205)	(N = 104)	(N = 20)	(N = 81)			
Average Age	35.47 (±12.60)	36.23 (±12.63)	32.35 (±8.53)	35.26 (±13.38)	*p* = 0.74		
Days with symptoms	1.60 (±1.85)	1.63 (±2.20)	1.85 (±1.18)	1.51 (±1.45)	*p* = 0.44		
Fever	33% (67/205)	37% (38/104)	35% (7/20)	27% (22/81)	*p* = 0.39	0.93	0.34–2.54
Sudden onset of symptoms	15% (30/205)	14.4% (15/104)	10.0% (2/20)	20.3% (13/81)	*p* = 0.78	0.65	0.13–3.13
Cough	65% (133/205)	71% (74/104)	70% (14/20)	56% (45/81)	*p* = 0.07	0.94	0.33–2.69
Odynophagia	33% (67/205)	37% (38/104)	35% (7/20)	27% (22/81)	*p* = 0.60	1.14	1.00–1.32
Dyspnea	5% (10/205)	5% (5/104)	10% (2/20)	4% (3/81)	*p* = 0.50	1.18	0.73–1.90
Irritability	9% (19/205)	8% (8/104)	10% (2/20)	11% (9/81)	*p* = 0.72	1.05	0.76–145
Diarrhea	9% (19/205)	6% (6/104)	5% (1/20)	15% (12/81)	*p* = 0.08	0.86	0.98–7.54
Chest Pain	9% (18/205)	4% (4/104)	25% (5/20)	11% (9/81)	*p* < 0.01	8.33	2.00–34.55
Chill	21% (43/205)	19% (20/104)	30% (6/20)	21% (17/81)	*p* = 0.55	1.8	0.61–5.22
Headache	62% (128/205)	61% (63/104)	55% (11/20)	67% (54/81)	*p* = 0.53	1.21	0.54–2.70
Myalgia	54% (111/205)	55.8% (58/104)	65% (13/20)	49% (40/81)	*p* = 0.40	1.47	0.54–3.99
Arthalgias	52% (107/205)	55% (57/104)	55% (11/20)	48% (39/81)	*p* = 0.64	1	0.38–2.63
Weakness	35% (72/205)	38% (39/104)	40% (8/20)	31% (25/81)	*p* = 0.57	1.1	0.41–2.95
Rhinorrhea	74% (154/205)	77% (80/104)	85% (17/20)	70% (57/81)	*p* = 0.33	1.7	0.45–6.29
Vomiting	5% (11/205)	4% (4/104)	15% (3/20)	5% (4/81)	*p* = 0.12	4.41	0.90–21.47
Abdominal pain	6% (12/205)	2% (2/104)	10% (2/20)	10% (8/81)	*p* = 0.05	5.66	0.74–42.84
Conjunctivitis	19% (38/205)	17% (18/104)	25% (5/20)	19% (15/81)	*p* = 0.72	1.59	0.51–4.49
	Outpatient treatment						
	Total	Patients with COVID 19	Patients with other viruses	Patients with negative PCR	*p*	OR	IC
	(N = 205)	(N = 104)	(N = 20)	(N = 81)			
Use of antibiotics prior to diagnosis	4% (8/205)	5% (5/104)	10% (2/20)	1% (1/81)	*p* = 0.69	1.42	0.11–9.42
Initiation of antiviral treatment	1% (3/205)	1% (1/104)	0% (0/20)	1% (2/81)	*p* = 0.62	0.83	0.77–0.90
Use of NSAIDs	42% (86/205)	47% (49/104)	40% (8/20)	36% (29/81)	*p* = 0.29	0.74	0.28–0.98

**Table 3 biomedicines-13-01012-t003:** Comparison between the performance of RT-PCR (reference method) and the PANBIO COVID-19 Ag RAPID TEST. NPV: negative predictive value; PPV: positive predictive value; LR(-): Likelihood ratio for a negative result. Kappa: 0.77.

PANBIO™ COVID-19 Ag RAPID TEST	Manual RT-PCR (Reference Method)		Sensitivity (%)	Specificity (%)	PPV (%)	NPV (%)	LR (-)
	Positive	Negative					
Positive	74	0	71	100	100	77	0.29
Negative	30	101					

PPV positive predictive value, NPV negative predictive value, LR likelihood ratio.

**Table 4 biomedicines-13-01012-t004:** Variation of diagnostic performance as a function of sampling time.

Performance Score by Days of Symptom Onset											
	N	TP	TN	FP	FN	PPV	NPV	Sensitivity	Specificity	LR	Index
								%	%	Negative	Kappa
Sampling at least 24 h after symptom onset	62	26	27	0	9	100%	75%	75%	100%	0.26	0.71
Sampling 1 day after symptom onset	57	20	26	0	11	100%	70%	65%	100%	0.35	0.62
Sampling 2 days after symptom onset	39	13	25	0	1	100%	96%	93%	100%	0.07	0.94
Sampling 3 days after symptom onset	27	8	15	0	4	100%	79%	67%	100%	0.33	0.69
Sampling 4 or more days after symptom onset	20	7	8	0	5	100%	62%	58%	100%	0	0.52

TP: True positive cases, TN: True negative cases, FP: False positive cases, FN: False negative cases, PPV: Positive predictive value, NPV: Negative predictive value, LR: Likelihood ratio.

**Table 5 biomedicines-13-01012-t005:** Median TCs by the reference method in positive and negative samples by PANBIO™ COVID-19 Ag RAPID TEST.

	PANBIO™ COVID-19 Ag RAPID TEST Positive	PANBIO™ COVID-19 Ag RAPID TEST Negative	Value of *p*
	Ct ± SD	Ct ± SD	
*Gene N*	17.54 ± 2.03	27.83 ± 5.87	*p* < 0.01
*Gene E*	16.70 ± 1.85	26.26 ± 6.93	*p* = 0.01
*Gene RdRP*	18.89 ± 2.46	27.50 ± 7.91	*p* = 0.03

**Table 6 biomedicines-13-01012-t006:** Characteristics of patients with false negative tests with PANBIO™ COVID-19 Ag RAPID TEST.

	Total	PANBIO™ COVID-19 Ag RAPID TEST True Positive	PANBIO™ COVID-19 Ag RAPID TEST False-Negative	*p*	OR	IC
	(N = 104)	(N = 74)	(N = 30)			
Average Age	35.91 (± 12.31)	35.17 (± 11.77)	37.73 (± 13.58)	*p* = 0.09	35.91 (± 12.31)	35.17 (± 11.77)
Days with symptoms	1.63 (± 2.20)	1.38 (± 1.46)	2.27 (± 3.36)	*p* = 0.44	1.63 (± 2.20)	1.38 (± 1.46)
Female	62.5% (65/104)	55.4% (41/74)	80% (24/30)	*p* = 0.02	62.5% (65/104)	55.4% (41/74)
Health workers	44.2% (46/104)	48.6% (36/74)	33.3% (10/30)	*p* = 0.11	44.2% (46/104)	48.6% (36/74)
Over 30 years	51.4% (53/104)	48.6% (36/74)	57% (17/30)	*p* = 0.52	51.4% (53/104)	48.6% (36/74)

**Table 7 biomedicines-13-01012-t007:** Multivariate regression analysis of people with true positive and false negative tests.

	Gl		F	P
Gender	1	0.509	1.554	0.411
Over 30 years	1	0.162	0.826	0.509

## Data Availability

The original contributions presented in this study are included in the article. Further inquiries can be directed to the corresponding author.

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
