# Peer review of "Detection of SARS-CoV-2 Using the Abbott™ PANBIO™ COVID-19 SELF-TEST Rapid Test in Patients Seen at INER"

_biomedicines, 2025, doi:10.3390/biomedicines13051012_

Round 1
Reviewer 1 Report
Comments and Suggestions for Authors
After reading the article carefully, I came to the following conclusion:
1. The article's assessment of self-testing instructions and observer bias is inadequate and needs to be further clarified and assessed whether the presence of an observer leads to behavioural biases (e.g. stricter compliance).
2. The authors are recommended to perform subgroup analysis (age, sex, symptom severity) and multivariable regression to identify clinically relevant factors that influence test sensitivity.
3. It is equally important to discuss the risk of cross-reactivity with MERS, SARS and OC43, despite the claimed 100% specificity, in order not to investigate the impact of the SARS-CoV-2 variant on sensitivity.
4. Insufficient statistical rigour: Clarify the methodology of Table 2 (e.g. adjustments for multiple comparisons). Provide confidence intervals for sensitivity/specificity. Replace ambiguous terminology (e.g. "general health damage" instead of "attacks on national generals").
5. The authors do not compare results with existing self-testing studies and should emphasise significant new findings/methodological improvements.
Author Response
Thank you very much for your comments, responses, and corrections to the text.
Comments and Suggestions for Authors
After reading the article carefully, I came to the following conclusion:
- The article's assessment of self-testing instructions and observer bias is inadequate and needs to be further clarified and assessed whether the presence of an observer leads to behavioral biases (e.g. stricter compliance).
It is a factor that cannot be determined in the discussion and was added as a great limitation that we did not consider.
- The authors are recommended to perform subgroup analysis (age, sex, symptom severity) and multivariate regression to identify clinically relevant factors that influence test sensitivity.
An analysis was carried out to evaluate whether demographic factors could interfere or increase the risk of false negatives, it was added in the results part.
- It is equally important to discuss the risk of cross-reactivity with MERS, SARS and OC43, despite the claimed 100% specificity, in order not to investigate the impact of the SARS-CoV-2 variant on sensitivity.
No information was found on the risk of false-positive results with other respiratory viruses; however, in our study, 6 people with seasonal coronaviruses were identified, and no false-positive antigen test results were found in patients with positive PCR result for other coronaviruses.
Added as part of the discussion.
- Insufficient statistical rigor: Clarify the methodology of Table 2 (e.g., adjustments for multiple comparisons). Provide confidence intervals for sensitivity/specificity. Replace ambiguous terminology (e.g., "general health damage" instead of "attacks on national generals").
The modifications have been made, and we have added the information requested.
Reviewer 2 Report
Comments and Suggestions for Authors
The paper by Canton-Cruz and colleagues is about the evaluation of the performance of a commonly used SARS-COV-2 rapid antigen assay.
The introduction is short, and it is not extremely informative. The methods section is written in a completely odd standard, and needs to be thoroughly revised.
The results section is complexively sufficient, a part from minor points. The discussion is lacking a sufficient effort of comparative analysis with respect to other similar studies, only a handful of which are cited,
while the amount of studies dealing with this specific assay is huge.
This poses a general major concern: it is not very clear to this reviewer what is the actual added value of this study in comparison to the long list of previous ones. This study doesn't deal with large cohorts, and is completely
lacking a-symptomatic subjects. As is also confirmed by the authors, this peculiar assay's sensitivity is proportional to the amount of viral load, (the paper shows that it tends to provide false negatives even at 20 Ct of RT-PCR).
This means that in a 'general population' screening, including a-symptomatics, the performance would have been probably very different, as reported in all studies designed including them (also in studies involving some of this paper's authors, and strangely not cited here).
In summary, I'm afraid that this study has a lower-than-required level of originality, and adds little to the current knowledge of this topic, since the assay it evaluates has been already widely analyzed, and often with better study designs. I strongly encourage the authors to consider some
expansion analysis, or joining forces with other datasets, or whatever else could empower this study, because in the current version, in my personal opinion, I'm not sure it can be worth a publication
Some minor points:
- line 8: please add address of institution (at least city and country)
- line 25: the COVID-19 health worldwide emergency actually begun in 2020
- line 29-260: please pay attention to reference numbers' styling, in the whole text
- line 32: ref 2 is from an online newspaper and looks inappropriate; please shift to a scientific publication or omit
- line 47: please add reference to this meta-analysis
- line 60: please specify which microbiology lab
- line 67: 'after consulting the kit guide'
- line 75: a) in this and in the following cases, in which a kind of a 'sub-chapter' section is created, with a separate title, it should be transformed to a proper
paragraph, with appropriate numeration and title styles
b) this specific section, with the full description of the test procedure (including illustrations, in spanish language) is absolutely inappropriate. The authors are invited to
change it into a synthetic description of the assay principles and features,including company name, and possibly a link to the protocol
- line 117, 123, 127, 134, 143: same as for line 75 a)
- line 119: please add referecne to these guidelines
- line 123: here, and in following text, please take care of verb tenses
- line 136: please rephrase the first sentence, and add kit and intruments' names, cat. numbers, and companies, as much as genral assay principles
- line 147-148: repetition of lines 144-145
- line 152: please rephrase 'negative symptoms'. Furthermore, the previous paragraphs seem to be stating that all involved subjects were symptomatic
- line 175: it is not very clear why such a large bias towards female subjects is present in the cohort
- line 179-181: from the text it is possible to understand that the 20 subjects with positivities for other viruses were all SARS-COV-2 negative. If this is correct, my question is: were the SARS-COV-2 positive
tested for the other viruses, as to assess possible co-infections (or assay cross-reactivity)?
- line 184: measure unit required for median symptoms onset
- line 189: "74 patients (or subjects) were positive"
- line 197: LOGIX kit is cited here, but the methods section state that a GENEFINDER kit has been used. Please clarify and correct
- line 204: the grey scale is not a good choice, since it is impossible to distinguish the different pathogens in the graph. Please change
- line 209: table 4 seems to have some spanish abbreviations. Furthermore, please explain non-obvious abbreviations in the table's footnotes
- line 212: figure 3 seems to be very low-res, and has some wording in spanish. Please re-plot in a new version
- line 215: in the discussion chapter, ref 12 and 13 are not found.
- line 255-256: please rephrase
- line 278: please check for correct authors names' style
- line 280: please note that 'CDC COVID-19 Surge Laboratory Group' is included in the 'et al.' indication
- line 287: please check for correct authors names' style
- line 290: please check for correct authors names' style
- line 292: please add link
- line 299, 302, 304, 306, 307: please check for correct authors names' style
- line 312: the full back matter section, including author contributions and funding, is missing
English is sufficiently correct in most of the paper's section, but I suggest a native speaker's revision, in particular to eliminate the chance of imperfect sentence structures reminescent of spanish language contructions, which can be found occasionally in text and figures.
Author Response
Thank you very much for your comments, responses, and corrections to the text..
Dear reviewer, we appreciate your comments and the time you have taken to review the article. We are sending responses to the comments and observations below.
The paper by Canton-Cruz and colleagues is about the evaluation of the performance of a commonly used SARS-COV-2 rapid antigen assay.
The introduction is short, and it is not extremely informative. The methods section is written in a completely odd standard and needs to be thoroughly revised.
Due to your suggestions, the introduction has been updated, in which we address in greater depth the importance of self-testing performed by each individual without the need to go to a laboratory.
The results section is complexively sufficient, apart from minor points. The discussion is lacking a sufficient effort of comparative analysis with respect to other similar studies, only a handful of which are cited, while the amount of studies dealing with this specific assay is huge.
We have revised the results section and made some corrections. Due to an error in the results entry, some incorrect numbers and terms had been entered, which have been corrected.
This poses a general major concern: it is not very clear to this reviewer what is the actual added value of this study in comparison to the long list of previous ones. This study doesn't deal with large cohorts, and is completely lacking a-symptomatic subjects. As is also confirmed by the authors, this peculiar assay's sensitivity is proportional to the amount of viral load, (the paper shows that it tends to provide false negatives even at 20 Ct of RT-PCR). This means that in a 'general population' screening, including a-symptomatics, the performance would have been probably very different, as reported in all studies designed including them (also in studies involving some of this paper's authors, and strangely not cited here).
Modifications have been made to the discussion. We have included comparisons with a recently published article that did not include studies validating the performance of the test used to prepare this article.
Perhaps one of the strengths was that the analysis was stratified by days of symptom onset, a result that supports the idea that the tests have the highest performance during the first few days. We also performed an analysis to assess whether the results were biased by certain sociodemographic factors of the subjects included, without any external factors affecting test performance. We did not include asymptomatic individuals, as the primary objective was to determine the performance of patients with mild respiratory illness.
In summary, I'm afraid that this study has a lower-than-required level of originality, and adds little to the current knowledge of this topic, since the assay it evaluates has been already widely analyzed, and often with better study designs. I strongly encourage the authors to consider some
expansion analysis, or joining forces with other datasets, or whatever else could empower this study, because in the current version, in my personal opinion, I'm not sure it can be worth a publication
Thank you for the comments. Adjustments have been made, and the importance of this review has been highlighted. As mentioned, there are systematic reviews. However, in the most recent one we reviewed, the test we evaluated is not included in the systematic review. There are few publications regarding the evaluation of the performance of the Abbott™ PANBIO™ COVID-19 SELF-TEST.
Some minor points:
- line 8: please add address of institution (at least city and country)
I added the institution's address: "Laboratory of Clinical Microbiology, National Institute of Respiratory Diseases (INER), Mexico City, Mexico."
- line 25: the COVID-19 health worldwide emergency actually begun in 2020
The year 2019 was corrected in the previous text, referring to the name of the virus. However, when correcting the introduction, it was reviewed to avoid confusion.
- line 29-260: please pay attention to reference numbers' styling, in the whole text
The references have been corrected. I hope there are no new errors.
- line 32: ref 2 is from an online newspaper and looks inappropriate; please shift to a scientific publication or omit
Reference was changed
- line 47: please add reference to this meta-analysis
Reference addedCai, P., Wang, J., Ye, P., Zhang, Y., Wang, M., Guo, R., & Zhao, H. (2024). Performance of self-performed SARS-CoV-2 rapid antigen test: a systematic review and meta-analysis. Frontiers in public health, 12, 1402949.
- line 60: please specify which microbiology lab
I specified the lab: "Clinical Microbiology Laboratory at INER, Mexico City, Mexico."
- line 67: 'after consulting the kit guide'
- line 75: a) in this and in the following cases, in which a kind of a 'sub-chapter' section is created, with a separate title, it should be transformed to a proper
paragraph, with appropriate numeration and title styles
b) this specific section, with the full description of the test procedure (including illustrations, in spanish language) is absolutely inappropriate. The authors are invited to
change it into a synthetic description of the assay principles and features,including company name, and possibly a link to the protocol
- line 117, 123, 127, 134, 143: same as for line 75 a)
- line 119: please add referecne to these guidelines
- line 123: here, and in following text, please take care of verb tenses
- line 136: please rephrase the first sentence, and add kit and intruments' names, cat. numbers, and companies, as much as genral assay principles
- line 147-148: repetition of lines 144-145
Complete correction was performed based on what was suggested about the methodology used to carry out the study.
- line 152: please rephrase 'negative symptoms'. Furthermore, the previous paragraphs seem to be stating that all involved subjects were symptomatic
Rephrase of "negative symptoms" implied in the rewriting of "A patient with respiratory symptoms" to "Patients with respiratory symptoms".
- line 175: it is not very clear why such a large bias towards female subjects is present in the cohort
The explanation about gender bias was added and an analysis was carried out in which no greater risk of obtaining false negative results based on gender was observed.
- line 179-181: from the text it is possible to understand that the 20 subjects with positivities for other viruses were all SARS-COV-2 negative. If this is correct, my question is: were the SARS-COV-2 positive
tested for the other viruses, as to assess possible co-infections (or assay cross-reactivity)?
I confirmed that the 20 with other viruses were SARS-CoV-2 negative; There were no co-infections documented in this study.
- line 184: measure unit required for median symptoms onset
I added unit "days".
- line 189: "74 patients (or subjects) were positive"
I corrected to "74 tests were positive."
- line 197: LOGIX kit is cited here, but the methods section state that a GENEFINDER kit has been used. Please clarify and correct
I corrected “LOGIX” to “GeneFinder™”
- line 204: the grey scale is not a good choice, since it is impossible to distinguish the different pathogens in the graph. Please change
The graph format was changed
- line 209: table 4 seems to have some spanish abbreviations. Furthermore, please explain non-obvious abbreviations in the table's footnotes
The changes were made based on your observations.
- line 212: figure 3 seems to be very low-res, and has some wording in spanish. Please re-plot in a new version
Improved image quality and made language adjustments
- line 215: in the discussion chapter, ref 12 and 13 are not found.
I corrected references 12 and 13 in the text.
- line 255-256: please rephrase
Rephrase de "non-precise" a "reliable".
- line 278: please check for correct authors names' style
It was corrected
- line 280: please note that 'CDC COVID-19 Surge Laboratory Group' is included in the 'et al.' indication
Was reviewed and modified
- line 287: please check for correct authors names' style
Was reviewed and modified
- line 290: please check for correct authors names' style
Was reviewed and modified
- line 292: please add link
Link added
- line 299, 302, 304, 306, 307: please check for correct authors names' style
Was reviewed and modified
- line 312: the full back matter section, including author contributions and funding, is missing
Added "Back Matter" section with contributions and funding.
Round 2
Reviewer 1 Report
Comments and Suggestions for Authors
The authors have addressed the last concern to a large extent with positive additional feedback, but the following minor issues remain and it is recommended that they be refined prior to formal acceptance:
1. Addition of functional validation experiments: for the N_323/K:53 mutation, it is suggested that reverse genetics or protein binding experiments be added to the discussion, or that the discussion include its possible effects on viral replication or host immune escape.
2. Evaluation of mutants should be expanded: If possible, the detection sensitivity data of the most commonly used mutations, such as XDV, JN.1.1 and XBB.1.5, BA.2.86, should be included.
3. Correct the logical inconsistency: no significant association (p = 0.816) with the variable "health care professional" in Table 7, and delete the speculative references to it (for example, "occupational effect") in the discussion so as not to mislead. 4.
4. Integration of the conclusion section: Emphasise the practical significance of the new data (e.g. Ct value, structural model) in the conclusion, e.g. "Mutations in high viral load regions may have an impact on PCR primer design, which should be monitored dynamically".
Author Response
Thank you very much for your comments. We believe they have certainly helped improve the quality of the text. Below are the latest observations and the changes we have made based on your suggestions:
- Addition of functional validation experiments: For the N_323/K:53 mutation, it is suggested that reverse genetics or protein binding experiments be added to the discussion, or that the discussion include its possible effects on viral replication or host immune escape.
Observations and results obtained by other authors when evaluating the N gene mutations were added to the discussion.
- Evaluation of mutants should be expanded: If possible, the detection sensitivity data of the most commonly used mutations, such as XDV, JN.1.1, XBB.1.5, and BA.2.86, should be included.
The findings regarding sensitivity and specificity with the predominant variants have been added to the discussion.
- Correct the logical inconsistency: no significant association (p = 0.816) with the variable "health care professional" in Table 7, and delete the speculative references to it (for example, "occupational effect") in the discussion to avoid misleading others. 4.
This parameter has been corrected and removed from Table 7.
- Integration of the conclusion section: Emphasize the practical significance of the new data (e.g., Ct value, structural model) in the conclusion, e.g., "Mutations in high viral load regions may have an impact on PCR primer design, which should be monitored dynamically."
The conclusions have been expanded and data added.
Reviewer 2 Report
Comments and Suggestions for Authors
I would like to thank the authors for their effort in modifying the paper after the reviewer's reccomendations. In particular I think the introduction and methods sections are really imrpved in the current form. The results section has been modified to a smaller extent, and probably some of the modifications are not real improvements.
The weak points about the lack of a-symptomatic subjects remains in place, and the real scientific significance of the study is still uncertain. The autohrs, correctly, state that this assay (PANBIO self- test) is widely used but poorly studied, so their work is original and interesting, but on the other hand, the strictly related (PANBIO rapid-assay), which shares good part of the examinated assay's features, has been extensively studied.
So, to try to improve the meningfulness of the article, in my opinion, the authors could:
- clearly introduce the 'ancestor' PANBIO rapid-assay, and briefly describe the differences and similiraties with the newer 'self-test' assay
- Expand, and organize more clearly their literature review, highlighting the papers which examinated the 'old' assay, and the new one (if any)
- Try to compare their findings with the 'old' Abbott assay, aong with similar non-abbott assays, in a clear and straightforward way.
- Discuss the limitations of not having a-symptomatic subjects, which is a serious limit to the possibility of establishing the 'real world' performance of the assay; also another limitation, of having a too low number of subjects with similar viruses (e.g. other coronaviruses) as to be able to really assess specificity
- Discuss more clearly the possible effects of some peculiar findings (i.e., the narrow time-frame for a optimal assay sensitivity; the strict correlation with RT-PCR Ct value). Finding that the assay is much more specific after 2 days post-symptoms onset is an interesting observation, but poses serious problems in real-word wide usage of the assay. Also, the fact that PCR positive samples with asl low as 20-21 Ct (which is a large viral load....) can provide false negatives is really a major pitfall of this assay. Asymptomatic subjects would probably have had, on average, higher Ct values, so, expectedly, higher proportions of false negative.
- All this said, the conclusions should draw the attention of the readers into for which possible applications this assay could still be recommended , and for which ones it would be better to use a different kind of assay.
A final consideration: the atuthors state that the 205 subjects were no other than symptomatic patients seeking for a diagnose in their lab. But, the new table 6 reveals that more or less on half of them were HCW, which is, let me say, at least a little bit odd.
Furthermore, this puts some arguments over the the assumption that this study is really representative of self-testing by non-experienced people. I would encourage the authors to try to provide some reasonable explanation to this strange finding, and to include a discussion of the possible futher limitations introduced by this bias.
Minor point: figure 3 and tab 4 have been modified, but not improved, since fig 3 still have poor resolution, and tab 4 have wrong titles for columns 2,3 and 4. Please correct
In general, I'll require a further round of revision to look for possible improvements of the paper following my more focused comments
Author Response
Thank you very much for your comments. We believe they have certainly helped improve the quality of the text. Below are the latest observations and the changes we have made based on your suggestions:
- Clearly introduces the 'ancestor' PANBIO rapid assay and briefly describes the differences and similarities with the newer 'self-test' assay.
The differences with the previous version have been explained, the most important being that the sample is obtained by untrained laboratory personnel and is performed and interpreted by the patient.
- Expand and organize more clearly their literature review, highlighting the papers that examined the 'old' assay and the new one (if any).
The bibliography has been expanded.
- Try to compare their findings with the 'old' Abbott assay, as well as with similar non-Abbott assays, in a clear and straightforward way.
It has been done, although the fundamentals of the test are the same, the performance of antigen tests has been affected by some mutations. Therefore, it is complex to compare the performance of the first trial with this latest version.
- Discuss the limitations of not having a-symptomatic subjects, which is a serious limit to the possibility of establishing the 'real world' performance of the assay; also another limitation, of having a too low number of subjects with similar viruses (e.g. other coronaviruses) as to be able to really assess specificity
Not having asymptomatic subjects has been identified as one of the important limitations.
- Discuss more clearly the possible effects of some peculiar findings (i.e., the narrow time-frame for an optimal assay sensitivity; the strict correlation with RT-PCR Ct value). Finding that the assay is much more specific after 2 days post-symptoms onset is an interesting observation, but poses serious problems in real-word wide usage of the assay. Also, the fact that PCR positive samples with asl low as 20-21 Ct (which is a large viral load....) can provide false negatives is really a major pitfall of this assay. Asymptomatic subjects would probably have had, on average, higher CT values, so, expectedly, higher proportions of false negatives.
Thanks for the comments. These have been mentioned as important limitations. Regarding CT values, studies have been included that demonstrate that CTs can be poorly compared, and that with variants that have been changing, this phenomenon has been observed: the CT threshold for obtaining better performance is lower compared to the first trials or with variants that have ceased to circulate. Although it is only mentioned that our data showed better performance at 48 hours, some authors have also obtained similar results.
- All this said, the conclusions should draw readers' attention to the possible applications for which this assay could still be recommended, and for which ones it would be better to use a different kind of assay.
The conclusions have been modified
- A final consideration: the authors state that the 205 subjects were no other than symptomatic patients seeking for a diagnosis in their lab. But, the new table 6 reveals that more or less on half of them were HCW, which is, let me say, at least a little bit odd. Furthermore, this puts some arguments over the assumption that this study is really representative of self-testing by non-experienced people. I would encourage the authors to try to provide some reasonable explanation to this strange finding, and to include a discussion of the possible future limitations introduced by this bias.
The data that were added to the results, analyzed to assess whether any factors or characteristics of the participants could influence the results, were obtained from the initial database. When evaluating participant characteristics, we considered it important that many of the participants, included, were healthcare personnel from the hospital itself. However, despite having knowledge of sample collection as nurses, doctors, and even inhalation therapy personnel, when performing a multivariate analysis, it was not observed that being personnel with prior knowledge of anatomy and perhaps having experience in sample collection influenced a lower number of false negative results. This has been explained in the discussion.
- Minor point: Figure 3 and Table 4 have been modified, but not improved, since Figure 3 still have poor resolution, and Table 4 have wrong titles for columns 2, 3, and 4. Please correct.
We corrected Table 4 and redid the graphs to try to obtain a higher-resolution image.
Round 3
Reviewer 2 Report
Comments and Suggestions for Authors
The paper has been further modifed, as to follow most of my recommendation from the second revision.
Just a few minor points still remain to be fixed:
- in general, literature about the Abbott assay hasn't been expanded to a sufficient extent, in my opinion
- line 42: ref 15 should now be re-numbered as ref 4, and all others should be renumbered in consequence
- line 212: figure 3 is not a higher resolution version of the previous one, it is indeed a completely different graph, with completely different values, so that it is impossible that it matches with table 5, which has remained the same as in the past version.... Also, in this new graph, it looks like, for E in particular, there is no way there can be any significant p values for the difference between positive and negative samples, as the two box-plots clearly have a large superimposable area. Please check thoroughly and correct what has to be corrected.
- lines 274-277 are written in spanish
- line 277: please correct 'faunding'
- line 282: this consideration is hard to be agreed. All exposed limitations affecting the reliability of Ct values are also affecting, and to a larger extent, antigenic tests. I would sincerely re-arrange the period dealing with the comparison with PCR results, which is presented as a topic of the paper, with a more analytic approach
- lines 392,397, 403, 408: pleas check for authors's names formatting
Author Response
Reviewer 2
We appreciate the time you have dedicated to the review, in response to the new comments we inform you:
- line 42: ref 15 should now be re-numbered as ref 4, and all others should be renumbered accordingly
We change the references and modify the progressive order.
- line 212: figure 3 is not a higher resolution version of the previous one, it is indeed a completely different graph, with completely different values, so that it is impossible that it matches with table 5, which has remained the same as in the past version.... Also, in this new graph, it looks like, for E, there is no way there can be any significant p values ​​for the difference between positive and negative samples, as the two box-plots clearly have a large superimpossible area. Please check thoroughly and correct what has to be corrected.
We understand, the graph has been modified, if we do not obtain the appropriate resolution, we could delete it. If you consider appropriate
- lines 274-277 are written in Spanish
I apologize for the serious error.
- line 277: please correct 'faunding'
The change is made
- line 282: this consideration is hard to be agreed. All exposed limitations affecting the reliability of Ct values ​​are also affecting, and to a larger extent, antigenic tests. I would sincerely re-arrange the period dealing with the comparison with PCR results, which is presented as a topic of the paper, with a more analytical approach
- lines 392,397, 403, 408: pleases check for authors' names format
It was reviewed